# Optimization of Processing Parameters and Adhesive Properties of Aluminum Oxide Thin-Film Transition Layer for Aluminum Substrate Thin-Film Sensor

**DOI:** 10.3390/mi13122115

**Published:** 2022-11-30

**Authors:** Yongjuan Zhao, Wenge Wu, Yunping Cheng, Wentao Yan

**Affiliations:** School of Mechanical Engineering, North University of China, Taiyuan 030051, China

**Keywords:** thin-film sensor, 1060 aluminum substrate, aluminum oxide thin film, process parameter optimization, performance testing

## Abstract

A thin-film strain micro-sensor is a cutting force sensor that can be integrated with tools. Its elastic substrate is an important intermediate to transfer the strain generated by the tools during cutting to the resistance-grid-sensitive layer. In this paper, 1060 aluminum is selected as the elastic substrate material and aluminum oxide thin film is selected as the transition layer between the aluminum substrate and the silicon nitride insulating layer. The Stoney correction formula applicable to the residual stress of the aluminum oxide film is derived, and the residual stress of the aluminum oxide film on the aluminum substrate is obtained. The influence of Sputtering pressure, argon flow and negative substrate bias process parameters on the surface quality and sputtering power of the aluminum oxide thin film is discussed. The relationship model between process parameters, surface roughness, and sputtering rate of thin films is established. The sputtering process parameters for preparing an aluminum oxide thin film are optimized. The micro-surface quality of the aluminum oxide thin film obtained before and after the optimization of the process parameters and the surface quality of Si_3_N_4_ thin film sputtered on alumina thin film before and after the optimization are compared. It is verified that the optimized process parameters of aluminum oxide film as a transition layer can improve the adhesion between the insulating-layer silicon nitride film and the aluminum substrate.

## 1. Introduction

As the development of the micro-machine field increases rapidly, the requirements for machining accuracy in machine manufacturing are also increasing. Cutting force, which is an important parameter in the metal-cutting process, directly affects the work-piece quality and tool life. As a result, it is particularly important to measure accurately the cutting force. In recent years, many researchers have conducted a great amount of work in cutting force [1,2,3,4]. Due to its small size and high precision, a film strain sensor can be used in embedded development and become one of the main directions of sensor development [5,6,7,8,9]. The 304 stainless steel is widely used as the substrate of thin-film strain sensors because of its low cost, easy polishing, and excellent mechanical properties [10,11]. In order to further improve the sensitivity of the thin-film strain sensor, a material which is more prone to elastic deformation than 304 stainless steel can be selected as the substrate to prepare the thin-film strain sensors [12,13].

A. Sanz-Herva studied the crystal quality, residual stress, and composition of aluminum nitride films with {0002} preferred orientation sputtered on metal surfaces (A10.9, Si0.1, Cr, Mo, and Ti). As a comparison, the values of strain and residual stress of aluminum nitride films deposited on metal surfaces are obviously lower than those of silicon oxide wafers [14]. Lorena Lopez prepared nanostructured titanium oxide films on various substrates. The growth, morphology, and thickness of titanium oxide micro-crystals changed significantly with different substrate materials [15]. P. R. Kishore Kumar et al. used RF magnetron-sputtering technology to prepare silicon nitride (Si_3_N_4_) films with different thicknesses on the surface of surface-treated and untreated aluminum alloys. It was found that the lattice strain of the films decreased with the increase in grain size. With the decrease in lattice strain, the lattice-matching rate and hardness increased [16]. Felipe C and others deposited titanium nitride films on brass and aluminum substrates through triode magnetron sputtering. By changing the bias potential and nitrogen flow rate (constant or variable) in the deposition process, the deposition characteristics of the films changed [17]. S. Thanikaikarasan et al. used constant-current electro-deposition technology to deposit Co-Ni (CoNi) thin films on stainless-steel (SS), aluminum (Al), tin oxide, copper (Cu), and other substrates [18], and found that the thin films deposited on the Cu substrates had definite physical, chemical, and magnetic properties. F.M. Mwema deposited aluminum thin films on glass and steel substrates at a constant substrate temperature of 90 °C and RF power of 350 W for 2 h. The topological structure and roughness characteristics of thin films on different substrates were studied. The results showed that the substrate type had an influence on the deposition, nucleation, growth, and film formation of aluminum thin films [19]. It can be seen from the above that researchers have studied aluminum oxide film, titanium oxide film, silicon nitride film, titanium nitride film, and other films, and have found that the performance of micro-films will vary with different conditions. At the same time, the properties of films deposited on different substrates have been studied.

In this paper, 1060 aluminum was selected as the metal substrate. The Stoney correction formula applicable to the residual stress of aluminum oxide film was derived theoretically. The influence of aluminum oxide film on the adhesion between the Si_3_N_4_ insulating layer and the aluminum substrate was tested. The preparation parameters of alumina thin films were optimized using an orthogonal test. 

## 2. Material Selection and Experimental Design

### 2.1. Selection of Substrate Materials for Thin-Film Micro-Sensors

The substrate of a thin-film sensor is used as a medium to transmit the strain of the tested component, and its ability to induce strain determines the sensitivity of the thin-film sensor. Theoretically, the more easily the substrate is strained, the higher the sensitivity of the thin-film sensor. Under the same stress on the substrate, the smaller the elastic modulus of the substrate material is, the greater the strain of the substrate is, as shown in Formula (1):(1)$$\varepsilon =\frac{\sigma }{E}$$
where σ represents stress (GPa), ε represents strain, and *E* represents the elastic modulus (GPa). Therefore, in order to enhance the sensor’s ability to sense strain, it is helpful to increase the sensitivity of thin-film sensors by selecting materials with a smaller elastic modulus than the substrate. The elastic modulus of 304 stainless steel is about 200 Gpa. Industrial pure aluminum (model 1060) can achieve the more effect through surface treatment, and it has a light and soft texture, excellent ductility, and corrosion resistance. It is often used to prepare gaskets, wire protective covers, and capacitors. Table 1 shows the comparison of mechanical properties between stainless steel and aluminum.

It can be seen from Table 1 that the elastic modulus and shear modulus of aluminum are about one-third of those of stainless steel, and its strain under the same stress condition is three times that of stainless steel. There is little difference between Poisson’s ratio and thermal expansion coefficient, so the yield strength of aluminum is low, but the stress caused by cutting force is very small, which will not reach the yield strength of aluminum.

In order to increase the application range of thin-film micro-sensors, different metals can be selected as substrates under different application conditions. In this paper, 1060 aluminum-H24 is selected as the metal substrate of thin-film micro-sensors for further study.

### 2.2. Theoretical Analysis of Bonding Performance between Alumina Film and Aluminum Substrate

Firstly, this paper studies the sputtering of a silicon nitride insulation layer directly on 1060 aluminum to isolate the sensitive layer film from the metal substrate. As an insulating layer, silicon nitride has an excellent insulating property, and the subsequent etching of the resistor gate will not damage it. In this paper, the silicon nitride thin film is prepared through ICP Deposition System SI 500D chemical vapor deposition, and its preparation process parameters are shown in Table 2. 

As shown in Figure 1, it was found that the sputtered silicon nitride film layer was uneven in color, and the film fell off, indicating that the adhesion between the two layers was not good. This is because the silicon nitride film was deposited in a high-temperature environment, and the thermal expansion coefficient of silicon nitride is 2.45 × 10^−6^/K, and that of 1060 aluminum is 23.6 × 10^−6^/K. The difference between the thermal expansion coefficients of the two materials is large, so the silicon nitride film prepared on the aluminum substrate cracked and fell off because of its poor adhesion. Therefore, if a transition layer is added between the aluminum substrate and the insulating layer, the difference between the thermal expansion coefficients of the two materials will be reduced, and the bonding performance of the two materials will be improved.

From Figure 1, the Si_3_N_4_ film on the 1060 aluminum substrate only fell off locally, but not in a large area. This was mainly because of the strong chemical activity of aluminum, and a dense layer of aluminum oxide film will be formed on its surface in the air, which shows that aluminum oxide has a good bonding performance with the silicon nitride film as a transition layer [20], and at the same time, aluminum oxide has a good material matching with the 1060 aluminum substrate, so aluminum oxide can be selected as the transition layer of the film sensor.

It can be known that the adhesive property between the insulating film and the metal substrate is closely related to the residual stress between them, and the residual stress mainly includes thermal stress and internal stress. Due to the difference in the thermal expansion coefficient between the film material and the base material, high sputtering temperature will produce greater thermal stress in the process of sputtering the insulating film. According to the theory of interface toughness proposed by A.A. Volinsky [21], Zhang Yuntao found that the film thermal stress had little effect on the interface toughness between the film and the substrate. It can be concluded from Formula (2) that the film thermal stress is mainly determined by the film’s own material characteristics and its thermal expansion coefficient with the substrate.
(2)$$\sigma_{t}=\frac{E_{f}}{1-v_{f}}\left(C_{f}-C_{s}\right)\left(T_{1}-T_{0}\right)$$
where *E_f_* is the elastic modulus of the film (GPa), *v_f_* is the Poisson’s ratio of the film, *C_f_* is the thermal expansion coefficient of the film (10^−6^/K), *C_s_* is the thermal expansion coefficient of the substrate (10^−6^/K), *T*_1_ is the sputtering temperature (°C), and *T*_0_ is the measurement temperature (°C).

The intrinsic stress of the thin film, that is, the internal stress, is mainly produced during the preparation process of sputtering deposition, which will change with the change in process parameters (such as power, gas flow rate, pressure, etc.), and is also closely related to the material of the thin film and the substrate. The processed substrate is absolutely flat in the initial condition. When the film grows on the substrate, the substrate will be slightly deflected under the action of the residual stress of the film. The curvature radius of the deflection can be measured by a laser interferometer and laser profile-meter, so the residual stress of the film can be characterized by the curvature radius of the substrate. Its principle diagram is shown in Figure 2.

The residual stress can be calculated by the Stoney formula, which needs to be applied under some assumptions: (1) the film thickness is much smaller than the substrate thickness; (2) the difference in the elastic modulus between the film and substrate is small; (3) the material of the substrate should be homogeneous and isotropic, and its initial state should be absolutely flat without deflection; (4) the film material is also isotropic; (5) the residual stress of the film is evenly distributed along the thickness direction; (6) the deformation of the film and the substrate is small and the edge of the film has little effect on the stress [22]. Sputtering silicon nitride insulating film on the aluminum substrate can satisfy the above assumptions. The residual stress in the film is determined by comparing the changes in the curvature radius of the substrate before and after growing the film, and Formula (3) is obtained.
(3)$$\sigma_{f}=\left(\frac{E_{s}}{1-v_{s}}\right)\frac{b_{s}{}^{2}}{6Rb_{f}}$$
in which *E*_s_ represents the elastic modulus (GPa) of the substrate, *v_s_* represents the Poisson’s ratio of the substrate, *b*_s_ represents the thickness of the substrate (mm), *b_f_* represents the thickness of the film (mm), and *R* represents the radius of curvature. Formula (2) shows that the thermal stress of the film has little effect on the interface toughness between the film and the substrate. Formula (3) shows that many physical properties of the substrate have an influence on the residual stress. Here, Formula (3) is used to calculate and analyze the bonding performance between the film and the substrate.

According to the research of the Stoney formula by X. C. Zhang et al., it was found that when the ratio of film thickness to substrate thickness is less than 0.01, the error of the result calculated by the Stoney formula is extremely small [23]. The thickness of the substrate of the thin-film sensor designed in this paper was 0.5–0.7 mm, the thickness of the alumina film was 800 nm, and the ratio of film thickness to substrate thickness was far less than 0.01. The classical Stoney formula is applicable to the case where the film thickness is much smaller than the substrate thickness and the elastic modulus of the film is close to that of the substrate. Because the difference between the elastic modulus of the base material and that of the alumina film is large, the measurement of film residual stress can be equivalent to the measurement of substrate curvature radius change; the classic Stoney formula revised by Shuai Hongjun and others from Shanghai University is adopted [24], as shown in Formula (4).
(4)$$\sigma_{f}=\left(\frac{E_{s}}{1-v_{s}}\right)\frac{b_{s}{}^{2}}{6b_{f}}\Delta \left(\frac{1}{R}\right)$$
where the curvature radius change is the curvature radius change caused by the deformation before and after the film is deposited on the substrate, and its expression is shown in Formula (5):(5)$$\Delta \left(\frac{1}{R}\right)=\frac{1}{R_{b}}-\frac{1}{R_{a}}$$

The change in curvature is characterized by the optical deflection method. The array laser is irradiated on the film, and the change in curvature radius is reflected by the displacement of the laser reflection point. The principle is shown in Figure 3:

Where *L* represents the relative distance (mm) between the sample and the collection point, *β* represents the incident angle of the laser beam, *t* represents the distance (mm) between the array laser beams, and Δ*t* represents the beam offset distance (mm).

Using *k* to represent the change in curvature radius, we can obtain Formula (6).
(6)$$k=\Delta \left(\frac{1}{R}\right)=\frac{\Delta t\cdot cos\left(\beta \right)}{2Lt}$$

To sum up, the revised Stoney formula is shown in Formula (7):(7)$$\{\begin{array}{l} \sigma_{f}=\left(\frac{E_{s}}{1-v_{s}}\right)\frac{b_{s}{}^{2}}{6b_{f}}k\\ k=\frac{\Delta t\cdot cos\left(\beta \right)}{2Lt} \end{array}$$
in which *v_s_* is the Poisson’s ratio of the substrate, *E_s_* is the elastic modulus (GPa) of the substrate, *h_f_* is the film thickness (mm), *h_s_* is the substrate thickness (mm), and *k* is the change in the surface curvature radius.

The beam offset distance Δ*t*, which is used to characterize the curvature radius, changes with the degree of film deflection, so assuming that the beam offset distance is a certain value, the surface curvature radius change *k* is a constant value. In addition, because the Poisson’s ratio of aluminum is very small, the influence of Poisson’s ratio can be ignored. The relationship between the residual stress *σ* of the alumina film and the elastic modulus *E* and thickness *b* of each substrate can be obtained by bringing the data of the mechanical properties of the substrate materials shown in Table 1 and those of the alumina film shown in Table 3 into the formula. As shown in Figure 4, it can be seen the residual stress of the aluminum oxide film on the aluminum substrate was 1.369 × 10^4^ GPa when the thickness of aluminum substrate was 0.7 mm.

### 2.3. Observation and Analysis of Surface Quality of Alumina Film

Alumina thin films were prepared using an FJL-560a magnetron and an ion-beam-sputtering deposition system on 1060 aluminum substrates. The sputtering gas was high-purity argon (99.99%), the reaction gas was high-purity oxygen (99.99%), and the sputtering target was high-purity aluminum with a purity of 99.999%. Firstly, the substrate was cleaned, and then sputtering was carried out under the process parameters of: argon flow rate of 50 sccm, oxygen flow rate of 1 sccm, power of 100 w, and pressure of 1 pa. A sample of alumina preparation on the aluminum substrate is shown in Figure 5; the size of the aluminum substrate is 30 × 20× 0.7 and the size of the aluminum oxide film is 26 × 20.

It can be seen from Figure 5 that the aluminum oxide film on the 1060 aluminum substrate was black in color, had a shiny surface, and excellent in effect. We followed the following process: Measure the thickness of the aluminum oxide film with the KLA-TencorP7 step meter. Select three different test points from top to bottom at the edge of the film to measure the thickness three times, and then calculate the sputtering rate according to the ratio of thickness to sputtering time. As shown in Table 4, the thickness and sputtering rate of the aluminum oxide film on the 1060 aluminum substrate was about 96.6 nm/min.

It can be seen from Table 4 that the peak difference in the film thickness on the 1060 aluminum substrate was 61 nm. The extreme value of film thickness of the alumina film was small, which indicates that its flatness and uniformity were good.

An X-ray diffractometer was used for testing and analysis, and the phase of alumina films sputtered on the 1060 aluminum was studied using incident X-ray diffraction (GIXRD) with an angle of 0.5° between the incident X-ray and the film surface. Cu-Kα radiation was used for detection on a Bruker D8 diffractometer, and the results are shown in Figure 6.

It can be seen from Figure 6 that there was no obvious metal characteristic peak on the substrates of 1060 aluminum, and the aluminum oxide film was generally amorphous. For alumina films, only when the substrate temperature exceeds 500 °C and the deposited particle energy is high, can crystalline alumina films be obtained. The experimental substrate was in the state of natural sputtering-temperature rise, and the sputtering temperature was about 40 °C, far less than 500 °C, so it was reasonable to obtain an amorphous alumina film.

The surface morphology of the alumina film on the 1060 aluminum substrate observed by the confocal microscope OLS5000 is shown in Figure 7.

It can be seen from Figure 7 that the surface quality of each point of the aluminum oxide film was relatively uniform. Although it can be seen that there were some micro-grooves on the surface of the aluminum oxide film, the quality of the surface roughness of the aluminum substrate was better than that of the silicon nitride insulating layer directly sputtered on the aluminum substrate. Therefore, the better the surface performance of the aluminum oxide film, the better the quality of the silicon nitride film of the subsequent insulating layer. The surface properties of alumina film have a crucial influence on the adhesion between the substrate and the film. In order to obtain better-quality alumina films on a 1060 aluminum substrate, the process parameters of preparing alumina films need to be optimized.

### 2.4. Orthogonal Experimental Design for Preparing Alumina Film on Aluminum Substrate

According to the research in the references, it was found that the sputtering pressure and substrate bias voltage affect the density of thin films, thus affecting the roughness of thin films. Therefore, the orthogonal test was designed according to sputtering pressure, argon flow rate, and substrate bias voltage in the film preparation process. Table 5 shows the level table of the orthogonal test factors.

Under the condition of sputtering power of 100 W and sputtering time of 450 s, three factors (sputtering pressure, argon flow rate, and substrate bias voltage) were tested by an orthogonal experiment. The experimental design and measurement results of the surface roughness of the alumina film are shown in Table 6.

## 3. Results and Analysis

### 3.1. Effect of Process Parameters on Surface Roughness and Sputtering Power of Thin Films

The influence degree of the above factors on the film roughness was analyzed using the range method. As shown in Table 7, R_j_ (j = A, B, C) in the table is the extreme range of roughness obtained by various factors, and K_i_ (i = 1, 2, 3) is the average value of measurement results at various levels.

Table 7 shows the influence of the sputtering pressure (A), argon flow rate (B), and substrate negative bias voltage (C) on the roughness of the alumina film was as follows: substrate negative bias voltage (C) > sputtering pressure (A) > argon flow rate (B). According to the value of k, the optimum technological parameters were A1-B2-C3. From Table 6, it can be found that when the negative substrate bias voltage changed from 0 to −30 or −40, the roughness of the film surface gradually decreased, which shows that the negative substrate bias voltage can improve the uniformity of the film surface. With the increase in sputtering pressure, the surface roughness of the film gradually increased.

Because changing some process parameters, such as sputtering pressure, argon flow rate, and substrate bias voltage, will affect the sputtering rate of thin films, and the sputtering rate will affect the preparation efficiency of thin films, it is necessary to study the influence degree of process parameters on the sputtering rate of thin films.

We measured the thickness of three points in the aluminum oxide thin-film sample with a KLA-Tencor step meter, and calculated the sputtering rate. The results are shown in Table 8.

From Table 8, regarding the peak difference of the alumina oxide thin film prepared on the 1060 aluminum substrate, the maximum value was that the peak difference of the thin-film was larger than 100 nm when the substrate negative bias voltage (test no.6 and 8) was zero, but the peak difference was 65 nm when the argon flow rate was 50 sccm, the sputtering power was 0.8 Pa, and the negative bias voltage was zero (test no.1), which further shows that the negative bias voltage had the greatest influence on the roughness of the thin film; with the increase in sputtering pressure and argon flow rate, the surface roughness of the films also increased.

Table 9 shows the extreme range analysis of the above three factors on the sputtering rate of the alumina film. In the table, V_j_ (j = A, B, C) is the extreme range obtained by each factor, and K_i_ (i = 1, 2, 3) is the average of the measurement results of each level.

It can be seen from Table 9 that the influence degree of sputtering pressure (A), argon flow rate (B), and substrate bias voltage (C) on the sputtering rate of the alumina film was as follows: sputtering pressure (A) > argon flow rate (B) > substrate bias voltage (C), and the optimum process parameter was A1-B2-C2 according to the value of K.

### 3.2. Surface-Fitting Analysis of Preparation Process Parameters of Alumina Thin Film

According to the extreme value of film roughness in Table 7, the main influencing factors of film roughness can be selected as sputtering pressure and substrate bias voltage, and the relationship model between the main influencing factors and film roughness can be established. Figure 8 shows the relationship model between process parameters obtained from the fitting analysis and film surface roughness; the relationship is shown in Formula (8):(8)$$Z_{1}=11.23+104.1x_{1}-0.853y-43.7x_{1}{}^{2}+0.473x_{1}\cdot y_{1}-0.00312y_{1}{}^{2}$$
where *Z*_1_ is the film roughness index (nm), *x*_1_ is the sputtering pressure (Pa), and *y*_1_ is the substrate bias voltage (V).

It can be seen from Figure 8 that with the increase in sputtering pressure, the scattering and collision of particles were intensified, which led to the non-uniformity of the sputtered film, and the defects on the film surface were enlarged, thus increasing the roughness of the film surface. According to the analysis results of the surface roughness range, with the increase in negative bias voltage of the substrate, high-energy electrons and ions are re-deposited. The collision between these particles and the film being formed can gain greater atomic momentum, thus increasing its mobility, which can increase the probability of deposition in a stable position and then form a dense film, and the surface roughness of the film decreases. From the trend in Figure 8, it can be seen that the preparation process parameters of film roughness were the minimum sputtering pressure (A1) and the maximum substrate bias voltage (C3).

According to the extreme value of film-sputtering rate in Table 9, it can be known that sputtering pressure and argon flow rate are the main factors affecting film-sputtering rate. The relationship model between the main influencing factors and film-sputtering rate is established. Figure 9 shows the relationship model between process parameters obtained by fitting analysis and film-sputtering rate, and the relationship between them is shown in the following Formula (9):(9)$$Z_{2}=93.78-9.797x_{2}+1.236y_{2}+3.051x_{2}{}^{2}-0.3567x_{2}\cdot y_{2}-1.445y_{2}{}^{2}$$

In the formula: *Z*_2_ is the sputtering rate index (nm/min), *x*_2_ is the sputtering pressure (Pa), and *y*_2_ is the argon flow rate (sccm).

As can be seen from Figure 9, two sputtering rates of 105.33 and 117.73 nm/min with test numbers 1 and 2 were out of the fitted surface, which may be due to the mismatch between the process parameters and sputtering power when preparing the thin film, so they did not conform to the rules of the relational model. With the increase in sputtering pressure, the ionization degree of gas increases, but the average free path of sputtered ions decreases, so it takes many collisions to reach the substrate, and the kinetic energy loss is too large, resulting in too small a migration distance when the sputtered ions reach the substrate. When the argon flow rate is too low, the ions of the sputtering target are insufficient; when the argon flow rate is too high, the gas particles in the sputtering chamber surge, and the ions generated by sputtering collide violently with the gas particles, so that the number of sputtering ions falling on the substrate decreases, resulting in the decrease in the film thickness, that is, the decrease in sputtering rate. According to the trend in Figure 9, the preparation process parameters of the fastest film-sputtering rate were a sputtering pressure (A1) of 0.8 Pa and an argon flow rate (B2) of 50 sccm. Although both of them are out of the model in Figure 9, considering the two process parameters, the sputtering rate is very high, so they can be used.

### 3.3. Observation and Analysis of Surface Quantity of Alumina Film

As shown in Figure 10, the alumina film prepared by the process parameters before and after optimization was observed using a Bruker Dimension Icon atomic force microscope in the area of 5 μm × 5 μm. It can be seen that the surface flatness of the film before optimization was weak, with obvious peaks and valleys, which were arranged irregularly. The roughness of the alumina film in the area of 5 μm × 5 μm was 8.06 nm, and the surface of the optimized film had no obvious valleys, which were arranged relatively neatly. The surface of the film was relatively flat, and the roughness of the alumina film in the 5 μm × 5 μm area was 7.144 nm, which is 11.36% lower than that before optimization.

To sum up, the optimized process parameters were a sputtering pressure of 0.8 Pa, an argon flow rate of 55 sccm, and a substrate bias voltage of −40 V. The surface quality of the aluminum oxide film prepared on the aluminum substrate was improved, and the preparation of the insulating silicon nitride film continued.

### 3.4. Influence of Alumina Film on Silicon Nitride Film of Insulating Layer

Alumina thin films were prepared on a 1060 aluminum substrate by optimizing the process parameters, and then silicon nitride thin films were sputtered on it using an ICP Deposition System SI 500D chemical vapor deposition system. The preparation process parameters of the silicon nitride film are shown in Table 1, and the silicon nitride film prepared on the alumina film before and after optimization is shown in Figure 11. It can be seen that the upper and lower boundaries of the silicon nitride film prepared on the pre-optimized alumina film had different degrees of cracking and falling off, which affected the bonding performance of the silicon nitride film. The silicon nitride thin film prepared on the optimized alumina thin film had no obvious cracking and shedding, and the bonding property was higher than that before optimization.

## 4. Conclusions

In this paper, the bonding performance of an aluminum oxide film and aluminum substrate was theoretically analyzed. Through orthogonal tests on the process parameters of preparing the alumina film, the effects of the process parameters on the surface quality and sputtering power of alumina were discussed through range analysis and surface-fitting analysis.

The Stoney correction formula applicable to the residual stress of an alumina film was derived. It can be seen from the formula that the residual stress of the film increased with the increase in the elastic modulus of the substrate, and the residual stress of the film also increased with the increase in the thickness of the substrate. 

Orthogonal tests were carried out on the preparation process parameters of alumina thin films, such as sputtering pressure, argon flow rate, and substrate negative bias voltage, to obtain the film roughness and sputtering rate. Through range analysis, it was found that the influence of process parameters on the roughness of the alumina thin films was as follows: substrate bias voltage > sputtering pressure > argon flow rate. The influence of process parameters on the sputtering rate of the thin films was as follows: sputtering pressure > argon flow rate > substrate bias voltage.

Through range analysis and curved surface-fitting analysis, the optimized preparation process parameters of the aluminum oxide film were determined as a substrate bias voltage of −40 V, an argon flow rate of 55 sccm, and a sputtering pressure of 0.8 Pa. By comparing the roughness of the alumina film before and after the optimization of the process parameters, it was found that the roughness value reduced by 11.36%. At the same time, better bonding properties could be obtained by sputtering a silicon nitride insulating layer on the aluminum oxide film prepared by using the optimized process parameters.

Aluminum was used as the substrate, and aluminum oxide film was added as the transition layer between the silicon nitride insulating layer and the aluminum substrate, which improved the adhesion of the insulating layer and improved the quality of the subsequent resistance grid layers.

## Figures and Tables

**Figure 1 micromachines-13-02115-f001:**
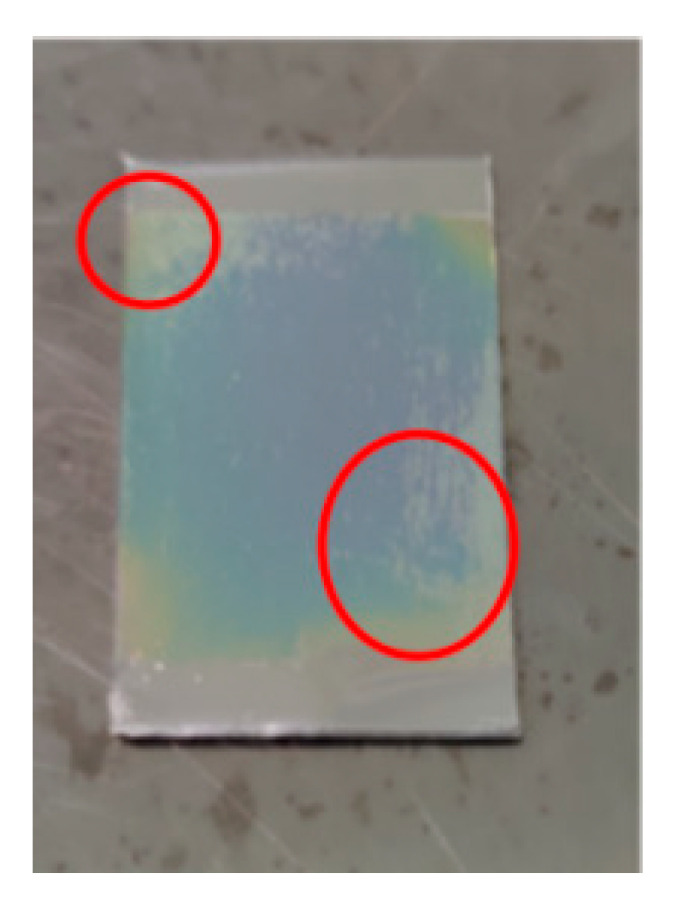
Sputtering silicon nitride film on substrate.

**Figure 2 micromachines-13-02115-f002:**
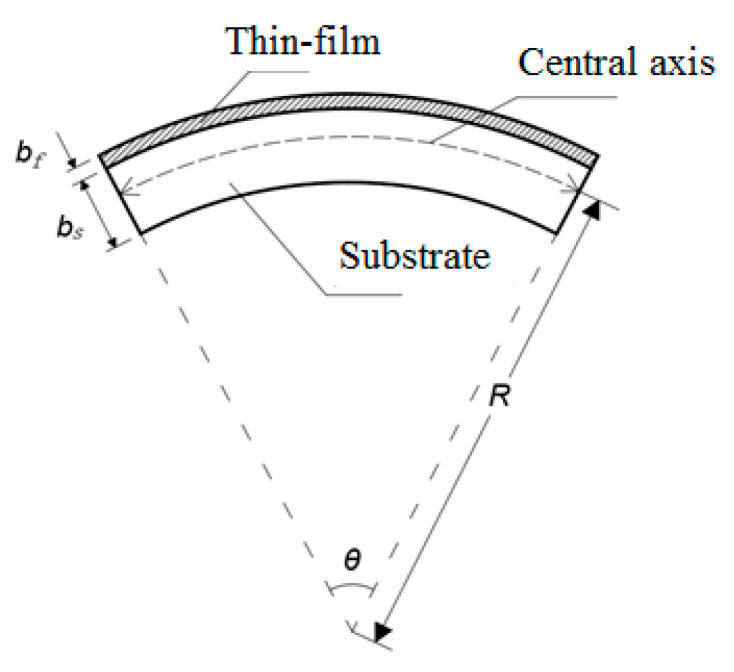
Schematic diagram of base deflection deformation.

**Figure 3 micromachines-13-02115-f003:**
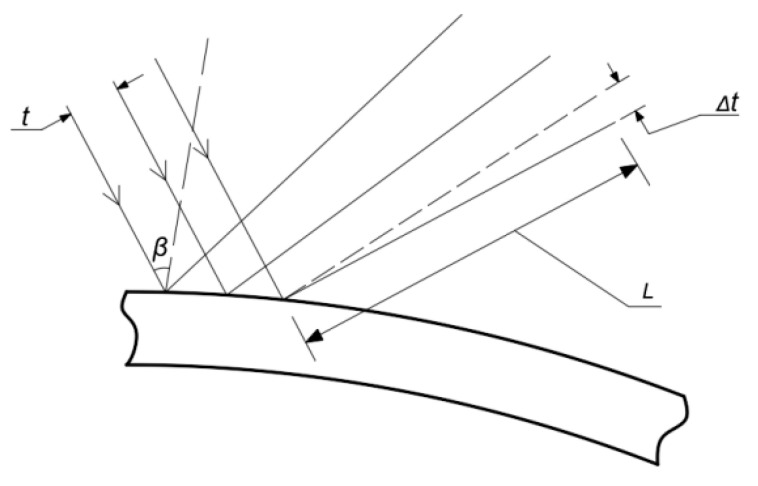
Schematic diagram of optical deflection method.

**Figure 4 micromachines-13-02115-f004:**
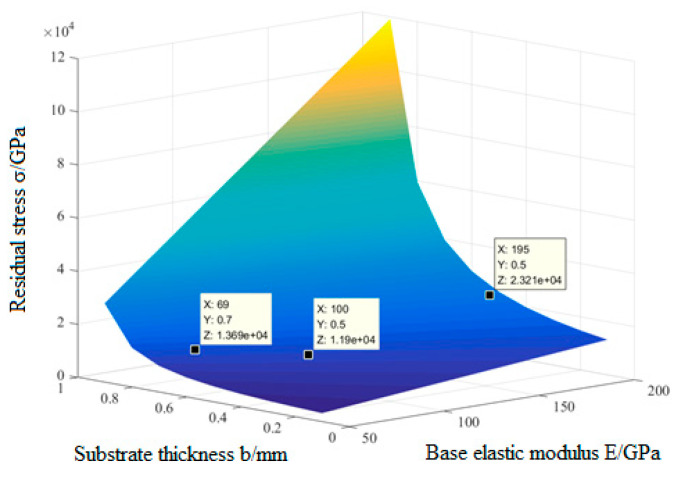
*b*, *E,* and *σ* relationship surface diagram.

**Figure 5 micromachines-13-02115-f005:**
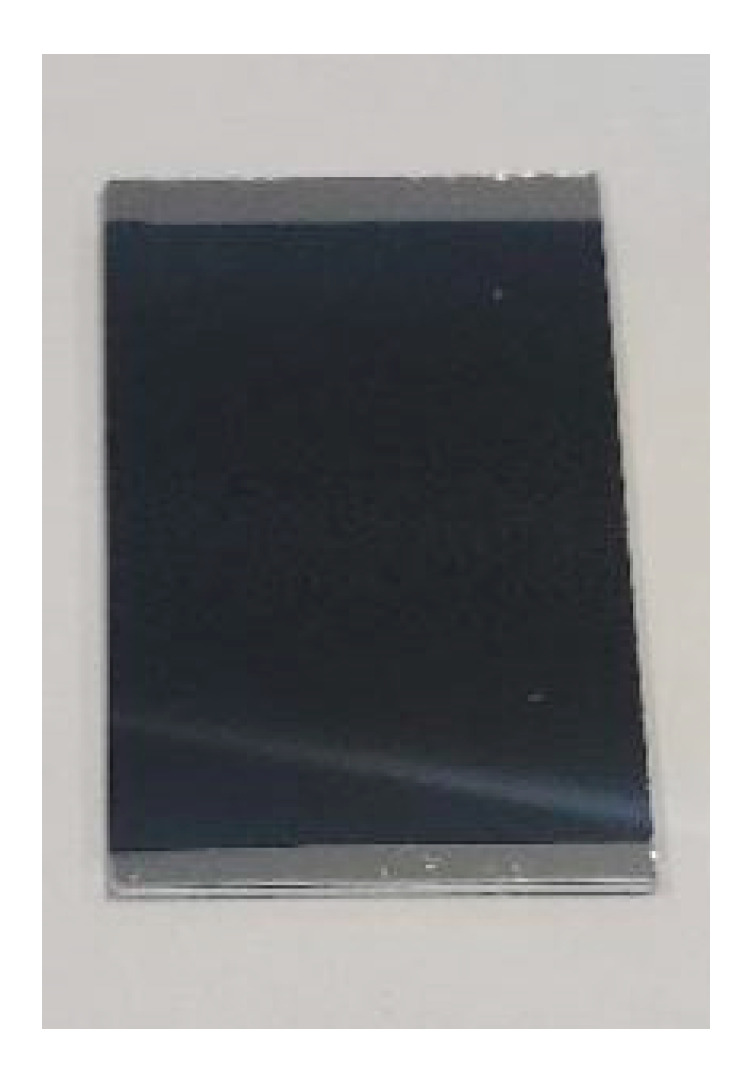
Alumina preparation sample.

**Figure 6 micromachines-13-02115-f006:**
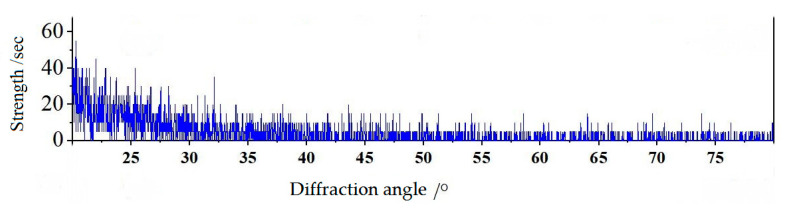
XRD pattern of alumina film on alumina substrate.

**Figure 7 micromachines-13-02115-f007:**
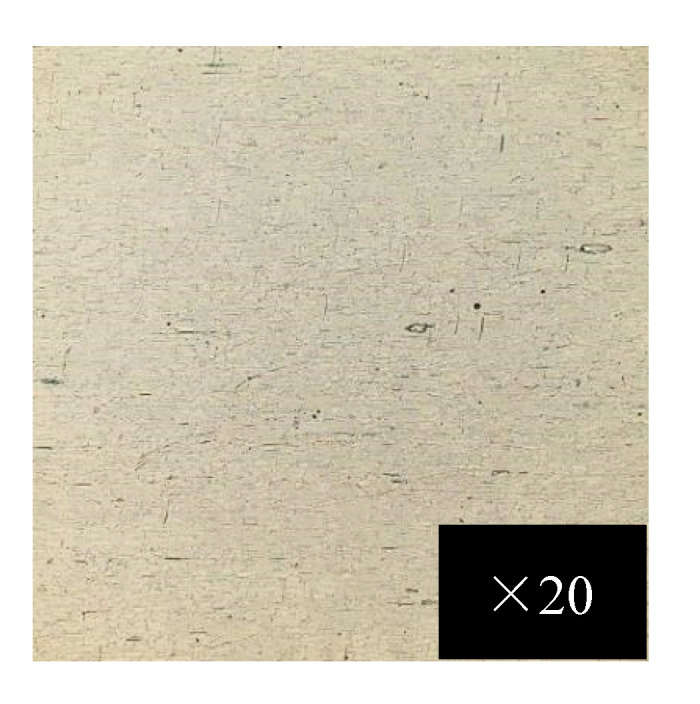
Alumina film on 1060 aluminum substrate observed by confocal microscope.

**Figure 8 micromachines-13-02115-f008:**
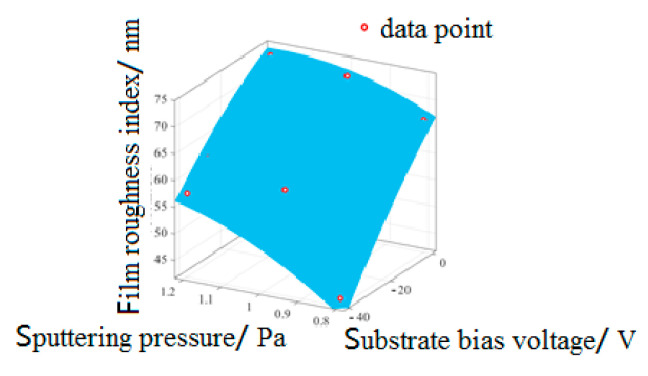
The relationship model between main influencing factors and film roughness.

**Figure 9 micromachines-13-02115-f009:**
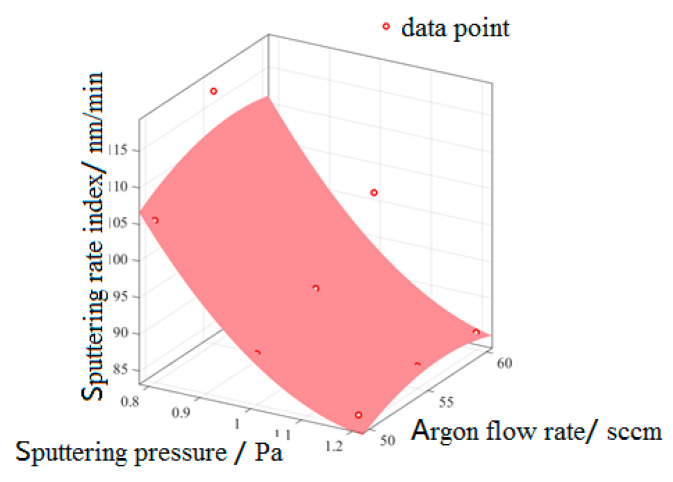
The relationship model between main influencing factors and film-sputtering rate.

**Figure 10 micromachines-13-02115-f010:**
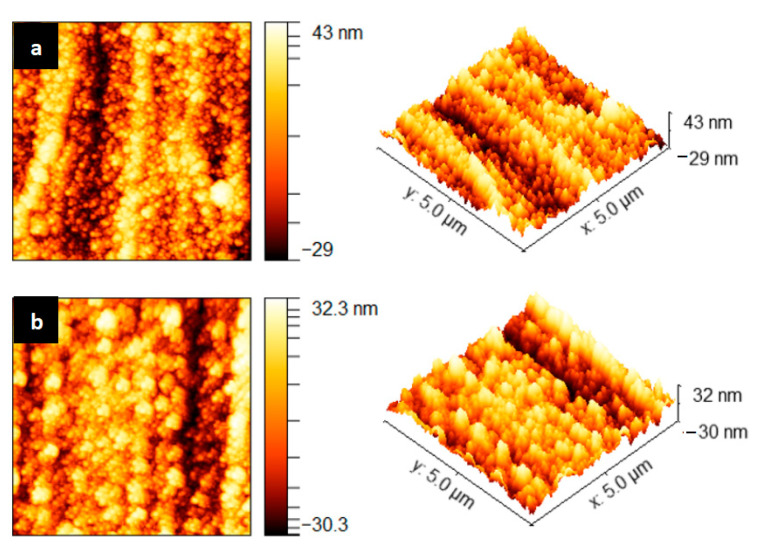
AFM comparison of optimization results of alumina film (**a**) Before optimization (**b**) After optimization.

**Figure 11 micromachines-13-02115-f011:**
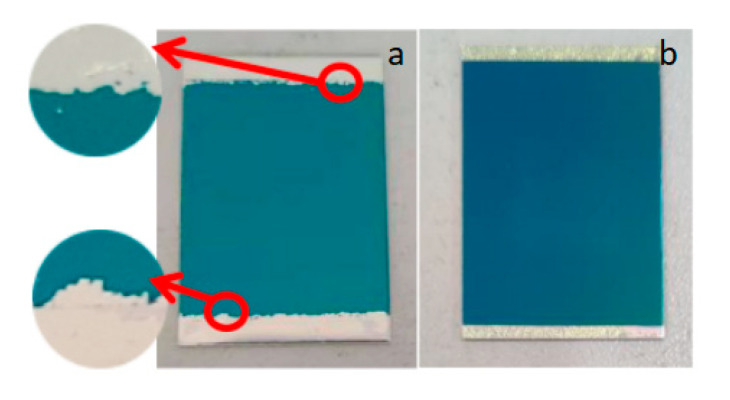
Comparison of Si_3_N_4_ films prepared by sputtering on aluminum oxide films before and after optimization. (**a**) Before optimization (**b**) After optimization.

**Table 1 micromachines-13-02115-t001:** Basic mechanical property parameters of base material.

Material	Modulus of Elasticity (GPa)	Shear Modulus (GPa)	Thermal Expansivity (10^−6^/K)	Yield Strength (MPa)	Poisson’s Ratio
304 stainless steel	195	75	18	205	0.29
1060 aluminum-H24	69	26	23.6	135	0.33

**Table 2 micromachines-13-02115-t002:** Process parameters of chemical vapor deposition silicon nitride film.

Film	Sputtering Power (W)	Sputtering Pressure (Pa)	Sputtering Temperature (°C)	Radio Frequency (MHz)	Ar_2_ Flow Rate (sccm)	SiH_4_ Flow Rate (sccm)	NH_3_ Flow Rate (sccm)
Si_3_N_4_	350	4	300	13.56	140	8	145

**Table 3 micromachines-13-02115-t003:** Basic mechanical property parameters of alumina film.

Thin-Film Material	Modulus of Elasticity/GPa	Shear Modulus/GPa	Thermal Expansivity/K	Poisson’s Ratio
Al_2_O_3_	357	143	(6.8−7.8) × 10^−6^	0.25

**Table 4 micromachines-13-02115-t004:** Thickness and sputtering rate of aluminum oxide film on 1060 aluminum substrate.

Substrate Material	1 (nm)	2 (nm)	3 (nm)	Average (nm)	Rate V (nm/min)
1060 aluminum	529	696	707	644	96.6

**Table 5 micromachines-13-02115-t005:** Orthogonal test factor level table for preparing alumina film.

Factor Level	A	B	C
	Sputtering Pressure (Pa)	Argon Flow Rate (sccm)	Basement Bias (V)
L1	0.8	50	0
L2	1	55	−30
L3	1.2	60	−40

**Table 6 micromachines-13-02115-t006:** Orthogonal experimental design and roughness results of aluminum oxide film prepared through magnetron sputtering.

Test Number	Sputtering Pressure (Pa)	Argon Flow Rate (sccm)	Basement Negative Bias (V)	Film Roughness Ra (nm)
1	0.8	50	0	66.389
2	0.8	55	−30	49.034
3	0.8	60	−40	43.156
4	1	50	−30	58.278
5	1	55	−40	50.261
6	1	60	0	71.920
7	1.2	50	−40	57.369
8	1.2	55	0	73.054
9	1.2	60	−30	61.427

**Table 7 micromachines-13-02115-t007:** Analysis of film roughness range.

	Film Roughness Ra/(nm)		
	A	B	C
K1	52.860	60.679	70.454
K2	60.153	57.449	56.246
K3	63.950	58.834	50.262
R_j_	11.09	3.23	20.192
	Influence degree: C > A > B		

**Table 8 micromachines-13-02115-t008:** Thickness and sputtering rate of aluminum oxide thin film on aluminum substrate.

Test Number	a (nm)	b (nm)	c (nm)	Average Thickness (nm)	Sputtering Rate (nm/min)	Peak Value Difference (nm)
1	825	785	760	790.00	105.33	65
2	862	785	867	838.00	117.73	82
3	734	767	806	769.00	102.53	72
4	723	669	637	676.33	90.18	85
5	688	677	745	703.33	93.78	68
6	830	735	721	762.00	101.60	109
7	584	674	652	636.67	84.89	90
8	672	724	793	647.00	86.27	121
9	633	684	608	641.67	85.56	76

**Table 9 micromachines-13-02115-t009:** Analysis of the extremely poor film-sputtering rate.

	Sputtering Rate V (nm/min)		
	A	B	C
K1	108.53	93.47	97.73
K2	95.19	99.26	97.82
K3	85.57	96.56	93.96
V_j_	22.96	5.79	3.86
	Influence degree: A > B > C		

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
