# Peer review of "Optimization of Processing Parameters and Adhesive Properties of Aluminum Oxide Thin-Film Transition Layer for Aluminum Substrate Thin-Film Sensor"

_micromachines, 2022, doi:10.3390/mi13122115_

Round 1

Reviewer 1 Report

The reviewer comments of the paper submitted to micromachines

“Study on Preparation Technology and Properties of Alumina Transition Layer on Aluminum Substrate of Thin-film Sensor”

The authors presented an article «Study on Preparation Technology and Properties of Alumina Transition Layer on Aluminum Substrate of Thin-film Sensor». The article is interesting and deserves the attention of readers. However, there are several points in the article that require further explanation.

Title needs to be concretized. What exactly is explored in the article? By what methods?

The abstract needs to be improved.

Demonstrate in the abstract novelty, practical significance. Add quantitative and qualitative work results to the abstract.

After analyzing the literature, show before formulating the goal of the "blank" spots. Which has not been previously done by other researchers? You must show the importance of the research being undertaken. Show what will be the new research approach in this article. You need to show a hypothesis.

In the last paragraph of the introduction, add scientific novelty and practical relevance. Add a clear purpose to the article.

Please consider the following papers:

“Experimental study and analysis of machinability characteristics of metal matrix composites during drilling”

“Investigation of progressive tool wear for determining of optimized machining parameters in turning”

Are all figures original? If not needed appropriate citations and permissions. Refine this for figures throughout the article.

Describe the measurement procedure in more detail. At what point in time? How is the measuring setup set up? How many repetitions of measurements? What statistical methods are used to process experimental results? Describe the experimental stand in more detail. What method of experiment planning is used and why?

Conclusions should be improved.

It is necessary to more clearly show the novelty of the article and the advantages of the proposed method. What is the difference from previous work in this area? Show practical relevance.

The article is interesting, but needs to be improved. Authors should carefully study the comments and make improvements to the article step by step.

Add 4-5 items of the findings of the study.

Reviewer 2 Report

In this paper, the authors have reported, optimization techniques to study the influence of process parameters on the surface quality and sputtering power of alumina by orthogonal test, and the range analysis and surface fitting analysis to improve the surface quality of alumina. The study shows promising results for the optimization process as revealed through the results and analysis section. However, there are several factors that should be incorporated before the manuscript can be accepted in its present format.

Abstract:

1. Why Si3N4 was used? Nothing is mentioned in the abstract about it..?

 Common errors in Manuscript:

1. Line 27 – “... measurement” should be written as “measure”.

2. Line 195: please indicate νs properly.

3. Error in sentence construction:  Line 62-63- ...... increase the adhesion between the substrate and the insulating layer silicon nitride, aluminium oxide was selected as the transition layer.

4. Error in sentence construction: Line 82 - The film sensor substrates studied previously is made of 304  stainless-steel, its elastic modulus is about 200Gpa.

5. Error in sentence construction: Line 302,303, 304: Measure the thickness of the thin-film sample obtained by orthogonal experiment, measure the thickness of three points in the aluminium oxide thin-film sample with P-7 step meter, and calculate the sputtering rate.

6. Error in sentence construction: Line 322-323: .......key parameters of process parameters, which is how to affect........

 Errors in References:

1. In the Introduction, use the references properly with only the surname of the author (no affiliations required) eg Sanz-Herva reported...... instead of  -  A. Sanz-Herva´s of Madrid Polytechnic University, Spain, studied.....

 2. Reference No 9  is not proper – Year is missing

 3. References are not properly formatted.

 4. Line 131-132: ...... Zhang Yuntao who is one of the research group..... . Such type of reference for a particular researcher should not be mentioned.

Figures:

1. Please edit the legend of figure 5.

2. Quality of figure 5 is very poor. please provide a good quality image with higher magnification (preferably as an inset) for better understanding. Also figure scale bar is missing.

3, Similarly, quality of figure 11 should also be improved. Scale bar for the figures are missing. Also higher magnification microscopic images would be better to visualize the changes clearly.

Technical Doubts:

1. In section 2.3 – line 234 indicates that sputtering rate of aluminum oxide film on 1060 aluminum substrate is about 99.8nm/min whereas the same thing is depicted as 96.6 in Table 4 which is a different value. Kindly confirm the same.

2. Microscopic observations of figure 7 indicated presence of grooves over the sputtered aluminium oxide. Did the author perform any SEM (scanning electron microscopy) image analysis to ascertain the same as the microscopic image is not very clear and also does not provide sufficient conclusive evidence. Moreover, did the authors perform any peel testing to ascertain sufficient adhesion of the sputtered aluminium oxide film over the two substrates?

 3. How was the thickness of deposited aluminium oxide film measured?  Was it done through scratch testing or in-situ measurement using a quartz crystal during deposition process?  Please clarify and mention in the manuscript.

4. AFM analysis to measure the surface roughness in figure 10 was performed over an area of 5 um x 5 um. Was the same scanning repeated at multiple sites before and after optimization process? Kindly comment on the same as the repeatability information is missing. Also, for roughness measurements, over large areas, a higher surface area (like 50 um x 50 um or more is better). Why have the authors not performed the same for their samples? 

5. In the conclusion section, points 1,2, and 3 need not be exclusively numbered. it can be written in separate paragraphs only.

Reviewer 3 Report

1. Question: We suggest that when expressing the research progress of other institutions, should not simply list them. Please summarize them. For example, we can't see your advantages when compared with the research of Madrid Polytechnic University.

2. Question: Please carefully check the correspondence between the drawing number and the figure mentioned in the article. Such asAs shown in Fig. 2, it can be found that the sputtered silicon nitride film layer is uneven in color, and the film falls off, indicating that the adhesion between the two layers is not good.

3. Question: Drawing in figure.1 is unclear and confusing, and there is no mark on the drawing. Please improve it.

4. Question: It is mentioned that the stress caused by cutting force is very small, which will not reach the yield strength of aluminum. How did you measure the stress caused by cutting force? Please prove it.

5. Question: Figure 5 is not clear. Whether the picture definition can be improved. The white dots in the picture will interfere with the reader's identification. Is the film falling off? Please explain and improve the picture.

6. Question: How did you measure and determine the sputtering temperature of 40 ℃? In addition, the text in Figure 6 is not clear, we suggest to use a more visual format of XRD.

7. Question: What equipment was used to observe the micro surface morphology of the film in Figure 7? Why did the article mention two different substrates, but only show the micro morphology of the aluminum substrate. We suggest standardizing the shooting of Figure 7, for example: ×20 is not standard, please improve it.

8. Question: In the abstract, you mentioned the surface micro morphology of aluminum oxide and silicon nitride before and after optimization is detected,’ we only noticed the microscopic morphology of alumina oxide, and there was no micro morphology of silicon nitride. Please explain it.

9. Question: You mentioned in the article that it is very important to accurately measure the cutting force of the tool, but we doubt whether your research in this paper can achieve accurate measurement. You did not mention this, and can you explain it.

10. Question: The references are not in uniform format. The author should improve according to the format requirements of the journal.

Round 2

Reviewer 3 Report

In this paper, the authors have reported, optimization techniques to study the influence of process parameters on the surface quality and sputtering power of alumina by orthogonal test, and the range analysis and surface fitting analysis to improve the surface quality of alumina. The study shows promising results for the optimization process as revealed through the results and analysis section. However, there are still few minor errors and queries that should be incorporated before the manuscript can be accepted in its present format.

Abstract:

1. Li ne 15-16 is incomplete…..

Figures:

1. Please edit the legend of figure 5. Name the three figures as Fig 5 a,b, and c and mention the difference between the three in the legend. Also figure scale bar is still missing which is very important and a standard way of representing any sample image. A simple high magnification optical microscopic image will be also good.

2. Scale bar for the figures in Figure 11 are still missing and has not been included.

3. Axis of Figure 6 should be in Only English and properly legible.... please correct it. Also why has the comparison with 304 Stainless steels been removed?

Technical Doubts:

1. In section 2.3 – line 229-230 still speaks about the.... “As shown in Table 4, the thickness and sputtering rate of aluminum oxide film on each substrate are as follows”.... This line indicates that a comparison is being made. Only the sputtering rate of aluminum oxide film on 1060 aluminum substrate was asked to be modified to the correct value. Then why did the authors suddenly remove the values of , Thickness and sputtering rate of aluminum oxide film on 3049 stainless-steel  from Table 4?

Similarly, the Legend on Table 4 also says “Table 4. Thickness and sputtering rate of aluminum oxide film on each substrate.”.... Kindly confirm the same and rectify it.

2. In the Microscopic observations of figure 7, incorporation of two images of similar magnification is of no use. Instead, the authors can incorporate the 2nd image of a higher magnification which will clarify the presence of grooves as explained.
